# The Dopaminergic Control of Movement-Evolutionary Considerations

**DOI:** 10.3390/ijms222011284

**Published:** 2021-10-19

**Authors:** Juan Pérez-Fernández, Marta Barandela, Cecilia Jiménez-López

**Affiliations:** Center for Biomedical Research (CINBIO), Neurocircuits Group, Department of Functional Biology and Health Sciences, Campus Universitario Lagoas, Marcosende, Universidade de Vigo, 36310 Vigo, Spain; martabarandela@gmail.com (M.B.); cecilia.jimenez.lopez@uvigo.es (C.J.-L.)

**Keywords:** basal ganglia, SNc, VTA, lamprey, salience, reward, aversion, posterior tuberculum, dopamine receptors

## Abstract

Dopamine is likely the most studied modulatory neurotransmitter, in great part due to characteristic motor deficits in Parkinson’s disease that arise after the degeneration of the dopaminergic neurons in the substantia nigra *pars compacta* (SNc). The SNc, together with the ventral tegmental area (VTA), play a key role modulating motor responses through the basal ganglia. In contrast to the large amount of existing literature addressing the mammalian dopaminergic system, comparatively little is known in other vertebrate groups. However, in the last several years, numerous studies have been carried out in basal vertebrates, allowing a better understanding of the evolution of the dopaminergic system, especially the SNc/VTA. We provide an overview of existing research in basal vertebrates, mainly focusing on lampreys, belonging to the oldest group of extant vertebrates. The lamprey dopaminergic system and its role in modulating motor responses have been characterized in significant detail, both anatomically and functionally, providing the basis for understanding the evolution of the SNc/VTA in vertebrates. When considered alongside results from other early vertebrates, data in lampreys show that the key role of the SNc/VTA dopaminergic neurons modulating motor responses through the basal ganglia was already well developed early in vertebrate evolution.

## 1. Introduction

The dopaminergic modulation from the substantia nigra *pars compacta*/ventral tegmental area (SNc/VTA) is critical for the control of movement. This is reflected in the severe motor deficits that appear when this dopaminergic system is compromised, as in Parkinson’s disease. Studies in lampreys and other vertebrates have shown that this system has an ancient origin, and is largely conserved through vertebrate evolution, which is reflected in the fact that motor deficits such as those seen in Parkinson’s disease appear in all vertebrate groups when SNc/VTA neurons are depleted [1,2]. Lampreys belong to the group of oldest living vertebrates that diverged from the line leading to mammals more than 500 million years ago, placing them in a key position for studying the origin and evolution of different neuronal circuits [3,4]. Numerous studies carried out over the last two decades have shown that the basal ganglia, together with the critical role of dopamine modulating the direct and indirect pathways in the striatum, are present in lampreys, together with many other features of the forebrain, showing striking similarities to mammals [5]. Striatal dopaminergic modulation arises from neurons in the posterior tuberculum (PT), which has consequently been proposed as homologous to the mammalian SNc/VTA [3,6,7,8]. Herein, we review the evidence supporting this homology, and discuss the evolution of the SNc/VTA and its role modulating motor responses with special reference to data available in lampreys and other key vertebrate groups [3,6,7,8].

## 2. Dopaminergic Populations and Receptors in the Lamprey

The first studies analyzing the dopaminergic system in non-mammalian vertebrates focused on the general distribution of dopamine neurons in the brain. Out of the two oldest groups of extant vertebrates, several studies have analyzed in detail the dopaminergic populations in different lamprey species [7,9,10], while currently no data exist regarding hagfish. Lamprey studies have been carried out using anti-dopamine and anti-tyrosine hydroxylase (TH) antibodies. TH is a reliable dopamine marker, given the lack of adrenaline and the scarce number of noradrenergic cells labeled by TH in these animals [9]. These studies have shown a pattern very similar to the dopaminergic populations found in other vertebrate groups (Figure 1a,b) [7,9,10,11]. From rostral to caudal, these populations are located in the olfactory bulbs (corresponding to the A16 group of mammals), preoptic region (resembling the suprachiasmatic nucleus population of other vertebrates) [7], hypothalamus (dorsal and ventral), nucleus of the postoptic commisure, mammillary area (suggested to be the homologue of the mammalian A11 population based on its projections to the spinal cord) [12], and the posterior tuberculum (homologue of the SNc/VTA; see below). Additionally, there are two populations in the lamprey rhombencephalon, in the isthmic region and close to the nucleus of the solitary tract. Additionally, there is a population of dopamine neurons in the spinal cord that co-express 5HT (Figure 1a) [13]. 

Moreover, the main dopamine receptors are present in lampreys, divided into two groups, one containing receptors that activate the adenylyl cyclase (D_1_-like family), and the other including those that inhibit it (D_2_-like family) [14]. This is not the only difference, since D_1_-like receptors are encoded by intronless genes, while those genes encoding the D_2_-like family receptors contain introns. Additionally, although both groups of receptors are coupled to G proteins, D_1_-like receptors are coupled to the Gs type, whereas D_2_-like family receptors are coupled to Gi/Go classes of the G protein. Regarding their structure, D_1_-like receptors show a short third cytoplasmatic loop and a long C-terminal tail, and D_2_-like receptors exhibit a long third cytoplasmic loop and a short C-terminal tail [15,16]. Thus, their ability to bind dopamine as a neurotransmitter is one of their few common features, and it has been proposed that this capacity was acquired by convergent evolution [17,18]. Several subtypes have been found within each family, and the repertoire of receptors varies depending on the vertebrate lineage (Figure 1c) as a result of whole genome duplications (WGDs) occurring in early vertebrate evolution [19], as well as lineage-specific gene losses. D_1A_ and D_1B_ are the only D_1_-like receptors present in mammals (also known as D_1_ and D_5_ in humans), whereas additional subtypes have been found in other vertebrates, namely, the D_1C_ and D_1D_ subtypes. The D_1C_ receptor was initially found in *Xenopus*, and later in some teleosts, and the D_1D_ receptor seems to be present exclusively in amniotes, although it is not present in mammals. A different receptor has also been found in carp, although its identity remains unclear and it was therefore named D_1X_. However, later phylogenetic studies renamed it as a subtype of D_1B_: D_1Bb_ [16,18,20,21,22]. Additional receptors belonging to the D_1_-like family, named D_1A1_ and D_1A2_, have been found in teleosts, thought to have arisen by the specific WGD that occurred in this lineage (3R; Figure 1c) [18,22]. Regarding the D_2_-like family receptors, three subtypes are present in most vertebrate groups (D_2_, D_3_, and D_4_), and additional subtypes have been reported in teleosts (D_2A_, D_2B_, D_4A_, and D_4B_), also stemming from 3R duplication (Figure 1c) [23].

Lampreys are thought to have diverged after the first WGD [24], but some data suggest that the branching occurred after the second WGD took place [25,26,27]. However, a recent study reconstructing the genome of early vertebrates suggests that lampreys diverged shortly after the first WGD [19]. The repertoire of dopamine receptors would agree with the divergence of the lamprey between the two WGDs, although it cannot be excluded that additional subtypes that might have arisen in the second WGD have been subsequently lost (Figure 1c) [22]. Only one receptor belonging to the D_1_-like family has been found in these animals [28,29], whereas two subtypes belonging to the D_2_-like family have been reported [3,22,30].

The receptor belonging to the D_1_-like family shows a high degree of similarity to the D_1A_ subtype, a broad expression pattern in the brain [29], and its functions in the striatum [28] and optic tectum/superior colliculus [8], which are largely conserved, strongly supporting its identity as a D_1A_ receptor. Within the D_2_-like family, one receptor shows clear phylogenetic similarities with the D_2_ subtype, with its brain-wide expression pattern and modulatory role in the striatum also supporting this homology [3,28,30]. Phylogenetic analyses place the other D_2_-like family receptor basal to the D_4_ sequences of other vertebrates, and its restricted expression pattern also supports this identity [22]. Given the high degree of conservation of the D_1A_ and D_2_ receptor sequences, their conserved expression areas, and that they are present in all vertebrate groups, it seems likely that these two subtypes carry out the more conserved functions, such as mediating the excitability changes in the striatum (see below). Consequently, they likely represent the ancestral D_1_-like and D_2_-like receptors from which the other subtypes arose through genomic duplications.

## 3. The Basal Ganglia Are Largely Conserved from Lampreys to Mammals

Although the basal ganglia are not the focus of this review, it is impossible to discuss the role of dopamine modulating movement without the basal ganglia framework. This group of subcortical nuclei plays a key role in selecting behaviors, motor control, and learning [5,31], and its origin in vertebrate evolution has remained a longstanding question. Work on the lamprey has shown that dopamine depletion results in severe motor deficits [2], suggesting that the basic role in mammals modulating motor responses is present also in these animals. Dopamine depletion using MPTP results in a marked hypokinesia, rigidity, and slower reaction times, thus resembling the deficits observed in Parkinson’s disease [2], which also appear in other vertebrates after dopamine is depleted [1]. A series of later studies showed that all basal ganglia components are present in the lamprey and that they show striking similarities with mammals in terms of intrinsic organization, inputs, and projections, as well as transmitters and synaptic and membrane properties [3,5,8,28,30,32,33,34,35,36,37].

In lampreys, as in mammals, the output of the basal ganglia is represented by the substantia nigra *pars reticulata* (SNr) and the globus pallidus *interna* (GPi) [5,33,34] (Figure 2a). These nuclei have tonically active inhibitory GABAergic neurons that project to different motor centers controlling various behaviors, such as locomotion or eye movements. The basal ganglia input nucleus, the striatum, is largely composed of GABAergic projection neurons (SPNs) divided in two types. One type of SPN expresses substance P and the D_1_ dopamine receptor, and dopamine increases the excitability of these neurons that directly target the output nuclei (SNr and GPi) through the direct pathway [28,33,38]. The other type of SPN expresses enkephalin and the D_2_ receptor, and its excitability is decreased by dopamine [28,30,33,38]. This subtype of SPN represents the indirect pathway, projecting to the inhibitory globus pallidus *externa* (GPe), which in turn targets the excitatory subthalamic nucleus (STN) [33]. The STN sends projections to the output nuclei, so that the net effect of this pathway results in an enhanced inhibition of the SNr/GPi, further increasing the inhibition of motor centers. Activation of the direct pathway, on the other hand, inhibits the output nuclei, facilitating actions (Figure 2a; see [5] for a review). Moreover, later studies identified the medial ganglionic eminence (MGE) in lampreys and hagfish [39,40]. The pallidum is known to originate from this region, and its presence therefore added developmental support to the lamprey basal ganglia.

The high degree of similarities between the lamprey’s forebrain and that of mammals is not restricted to the basal ganglia. Recently, a pallial region has been identified as a homologue of the dorsal pallium and thus the precursor of the mammalian neocortex [4,42,43,44]. The presence of cortical homologues in the lamprey [4,42,43,44] and other forebrain regions, including the thalamus and the habenular system (see [4]), indicate that not only the basal ganglia and the dopaminergic system, but the main areas and pathways conforming to the basic architecture of the forebrain were already present in early vertebrates.

## 4. The SNc/VTA at the Base of Vertebrate Evolution

As outlined above, the key role of dopamine modulating the striatal direct and indirect pathways is present in both lampreys and mammals. The lamprey dopaminergic population located in the posterior tuberculum (Figure 1a and Figure 2b) was first proposed to be the homologue of the mammalian SNc/VTA in the early 1970s [6]. The homology of the dopaminergic neurons in the PT with the SNc/VTA was initially controversial due to its position in the diencephalon. However, this dopaminergic nucleus was later shown to target the striatum [7], suggesting that its role modulating the input nucleus of the basal ganglia is also present in the lamprey. Later work confirmed this hypothesis when the direct and indirect pathways from the striatum were characterized [28,30]. As in mammals, striatal neurons expressing D_1_ or D_2_ receptors form separate populations in the lamprey that target the output nuclei, either directly or indirectly (Figure 2a), and their excitability is increased by dopamine in the case of D_1_-expressing neurons and decreased in the case of those expressing D_2_ [28].

Apart from striatal modulation, subsequent studies have revealed more similarities between the lamprey PT dopaminergic population and the mammalian SNc/VTA. First, the overall connectivity of the lamprey dopaminergic population shows striking similarities to that seen in mammals (Figure 2a) [3]. Apart from the projection to the striatum, reciprocal connections from striosomes have been found [3,36], elucidating the ancient origin of the nigrostriatal loop. Reciprocal connections have also been found between the SNc/VTA and STN, suggesting that like mammals, the SNc modulates this nucleus [45]. The SNc/VTA also receives sensory information from multiple areas. Olfactory information reaches the SNc/VTA from both the medial olfactory bulb [46], a projection shown to contribute to locomotion [47,48], and from the main olfactory bulb [49]. Visual inputs reach the SNc/VTA from the tectum (homologue of the mammalian superior colliculus; SC), as well as the pretectum [3,8,50]. The tectum/SC projection suggests that the same mechanisms thought to convey salience-related visual information to the SNc/VTA in mammals are also present in the lamprey (see below; [3,8]). Additionally, the SNc/VTA receives inputs from other sensory regions. This includes the octavolateral area [3], a region that integrates mechanosensory, electrosensory, and vestibular information [51,52], and the dorsal column nucleus, which receives mechanosensory information from the spinal cord. Other evolutionarily conserved inputs to the SNc/VTA include projections from the cortical homologue (pallium), habenula, thalamus, and pedunculopontine nucleus (Figure 2a) [3].

Much like in mammals, lamprey’s SNc/VTA dopaminergic neurons express D_2_ receptors [3]. Dopamine release in the SNc/VTA is controlled by D_2_ autoreceptors [53], and this mechanism is therefore very likely present in lampreys as well. The lamprey SNc/VTA population consists of around 600 dopaminergic neurons [41], a seemingly low number, but which is in line with the percentage of SNc/VTA neurons found in mammals, not taking the large cortical expansion into account in the latter [41,54]. Another key feature of mammalian SNc/VTA neurons is that they exhibit tonic activity that results in a baseline release of dopamine in the striatum [55], and SNc/VTA neurons in the lamprey are also tonically active [8]. Further similarities between the lamprey and mammalian SNc/VTA include co-expression of dopamine with other neurotransmitters (Figure 2b–d), the co-release of dopamine and glutamate in the striatum (Figure 2j,l,m) [41], and the habenular control of dopamine release [56]. Altogether, this shows that the lamprey’s dopaminergic population in the posterior tuberculum bears striking similarities to the mammalian SNc/VTA and should therefore be regarded as its homologue.

## 5. Early Origin of the Dopaminergic Coding of Salience and Reward

Although mostly known for coding reward, dopamine SNc/VTA neurons have a phasic two-component response that reflects salience/novelty and reward value, respectively (Figure 3a) [55,57,58,59,60,61]. The first component is independent of the sensory modality and its reward value, depending only on its salience (meaning here its physical intensity). The intensity determines the ability of a stimulus to capture our attention, and therefore salience may be best described as the ability to stand out from other stimuli based on simple features, such as size or speed. This short-latency dopaminergic signal is activated in a graded manner, with higher salience reflected by increasing amplitudes. It has been proposed that going from unselective event detection to value processing is performed via intermediate steps, which include the identification of the stimulus and its value. The processing involved in these sequential steps most likely takes place in several brain areas, although little is known about the mechanisms conveying this information to SNc/VTA neurons [57,58,59,62].

Dopaminergic SNc/VTA neurons have been shown to code salience also in lampreys (Figure 3c–e) [8]. Looming stimuli (an expanding dot presented in a screen mimicking an object approaching) evoke larger responses in dopaminergic neurons in parallel with the increase in visual speed (Figure 3c–e). As in mammals, these neurons consequently produce stronger responses to more intense stimuli [8]. As mentioned above, a direct projection from the tectum/SC to the SNc/VTA is also present in lampreys (Figure 2a and Figure 3b) [8]. The tectum/SC is a phylogenetically ancient structure that can produce gaze-shifting eye movements independently from other brain regions. Visual information is integrated according to a retinotopic map in the SC/tectum and activates neurons in the deep layer.

Neurons in the deep layer subsequently project contra- or ipsi-laterally to the brainstem to evoke orienting or evasive responses, respectively. Additionally, visual inputs activate interneurons that provide lateral feedforward inhibition that silences competing areas, which in turn allows for stimulus selection [51,63,64,65]. This architecture is very similar to that observed in the tectum/SC of other vertebrate groups, including mammals, indicating a large degree of conservation [66]. In mammals, the SC has been shown to directly project to the SNc/VTA and influence it in response to unexpected and salient visual stimuli [55,67,68,69,70]. This projection was also demonstrated to relay reward-predicting visual information to dopaminergic neurons [71].

Additionally, recent work in mice has shown that a direct projection from the SC to the SNc regulates locomotion-speed signals to dopamine neurons, and enhances appetitive locomotion [72]. Moreover, direct projections to the VTA have been related to wakefulness and innate defensive behaviors [73,74], indicating that the SC-SNc/VTA pathway plays several important roles. In the lamprey, ongoing work shows that the increase in activity parallel to stimulus intensity is abolished when the tectum is inactivated [75]. Such a relationship indicates that the pathway conveying salience information from the SC/tectum to the SNc/VTA has an ancient origin. Although the underlying mechanisms of salience coding are not fully understood, we may conclude that the signaling of alerting events represents an ancient and conserved ability of dopamine neurons.

Regarding the second component of the dopaminergic neurons’ response related to reward/aversion, it is now known that the lateral habenula also controls dopamine activity in lampreys (Figure 2a) [35,56]. In mammals, a tonically active population of neurons in the globus pallidus (GPh) projects to the habenula, and an increase in GPh activity in turn increases the excitatory input to the lateral habenula, resulting in inhibition of dopaminergic neurons. GPh neurons can code the value of actions. When the outcome of such an action is worse than the prediction, GPh activity increases, and further actions are discouraged. In contrast, when the outcome offers a higher reward than expected, the decrease in GPh activity results in an increase in dopamine activity, thus promoting the behavior [76,77]. The lateral habenula’s signaling of aversive behaviors is a feature that appeared early in vertebrate phylogeny, as the complete striosome–GPh–lateral habenula circuit is present in the lamprey, as originally described in these animals [35]. This structure provides direct excitation to dopamine neurons, as well as indirect inhibition via the GABAergic rostromedial mesopontine tegmental nucleus (RMTg) [35,56,78]. In the zebrafish, the lateral habenula plays a critical role in evaluating actions and providing behavioral flexibility [79,80]. However, no direct projections to the dopaminergic system have been found in teleosts [81,82], suggesting that this connection was secondarily lost in teleosts but maintained in the evolutionary line leading to mammals. Additional studies are necessary, however, to investigate whether the indirect control through the RMTg is present in teleosts.

Dopamine neurons thus code salience in lampreys, indicating that the short latency signal in the SNc/VTA, occurring prior to the identification of an object and its reward value, was consequently present already in early vertebrate evolution. Additionally, although functional studies are still needed, the high degree of development of the habenular system in lampreys suggests that at least some of the mechanisms underlying reward-related signals were present in the first vertebrates. There is therefore evidence that the dopaminergic system, together with the basal ganglia have been present throughout vertebrate phylogeny, having already evolved their basic functions at the beginning of vertebrate evolution.

## 6. Direct Dopaminergic Control of Downstream Centers

Apart from the well-described dopaminergic role modulating motor responses via projections to the striatum, direct modulation of motor centers in the midbrain and brainstem from SNc/VTA neurons has recently been described. These extrastriatal pathways include projections to the mesencephalic locomotor region (MLR), described in the lamprey, salamander, and rat [3,83,84,85], to reticulospinal neurons (lamprey) [86], and to the tectum/SC. This last projection has been functionally described in lampreys [3,8,41] and is also present in amphibians [87], zebrafish [88], and mammals [89]. Interestingly, the superior colliculus in mammals has been shown to receive additional dopaminergic inputs from the zona incerta [90] and locus coeruleus [91], suggesting a complex role of dopamine modulating this structure.

Interestingly, the same dopaminergic neurons have been shown to send axonal branches to both the striatum and tectum/SC, both in lampreys and mammals, and thus both structures become modulated in parallel [8,41,89]. Dopamine neurons projecting to both the striatum and MLR have also been described in lampreys and rats, where most neurons projecting to the striatum also target the MLR [84]. This scenario shows that motor centers are rapidly modulated or primed via the direct projections, while a slower modulation occurs via the basal ganglia. The presence of two parallel pathways, one quickly priming motor centers and the other providing a slower modulation through the basal ganglia, may be related to the previously outlined biphasic signaling components of SNc/VTA dopaminergic neurons. The existence of modulatory pathways bypassing the basal ganglia suggests that their role is to quickly enhance the responsiveness of motor centers to allow subsequent faster responses. Accordingly, it has been shown that a non-action specific increase in dopaminergic neuron activity can invigorate future actions [92]. Moreover, the net effect of dopamine in the lamprey tectum is to enhance visuomotor integration, giving rise to larger responses for the same visual inputs (Figure 3f) [8]. The dopaminergic modulation therefore allows for a more efficient preparatory process, optimizing behavioral responses by directly tuning the motor centers.

Altogether, one may conclude that the SNc/VTA modulation of motor responses is more refined than previously thought. Moreover, given the large degree of conservation of SNc/VTA neurons, as well as the presence of their direct pathways to motor centers across different vertebrates, it seems logical that direct modulation may have a role in motor deficits observed in Parkinson’s disease [8,85]. The dopaminergic fibers found in the pedunculopontine nucleus (PPN), which is part of the MLR [84], suggest that the same kind of direct dopaminergic modulation is also present in humans.

## 7. Co-Release of Dopamine and Other Neurotransmitters

Although the focus of studies on the SNc/VTA has been dopamine-related mechanisms, these nuclei are formed by heterogeneous populations that include several neurotransmitter phenotypes. Single-cell RNA sequencing has revealed that mesodiencephalic neurons can be divided into seven subgroups, five of which express dopaminergic markers, and two express glutamate and GABAergic markers [93]. GABA, glutamate, and dopamine are co-expressed in different combinations in the mammalian SNc/VTA, where non-dopaminergic neurons are also present [94,95,96,97,98,99]. The SNc and VTA show differences in the relative number of neurons belonging to each neurotransmitter phenotype. For instance, neurons co-expressing dopamine and glutamate are more abundant in the VTA than in the SNc [94,95,96,98]. Functionally speaking, the co-release of dopamine/glutamate and dopamine/GABA has been demonstrated in the mammalian striatum [100,101,102,103,104]. In addition, glutamatergic neurons have been shown to play a key role mediating innate defensive responses [105], which agrees with VTA being partially activated by aversive stimuli [106]. In general, the abolishment of SNc/VTA glutamate transmission will lead to a significant alteration of reward-related responses [107,108]. In zebrafish, dopamine is also co-expressed with glutamate and GABA in neurons of the posterior tuberculum [109]. No co-expression studies have been performed in other vertebrate groups. In lampreys, dopamine in the SNc/VTA is co-expressed with glutamate [41,110] and GABA [41]. Some neurons have also been found to co-express dopamine, glutamate, and GABA, although the most abundant phenotype was shown to be dopamine–glutamate co-expressing neurons, and a large proportion of SNc/VTA dopamine neurons co-expressing glutamate (Figure 2c) [41]. The neurotransmitter phenotypes in the lamprey are consequently the same found in mammals and zebrafish, indicating that the co-transmission of dopamine with other neurotransmitters is an ancient feature in vertebrate evolution. In agreement with the anatomical data, glutamate and dopamine from the SNc/VTA have been shown to be co-released in the MLR and striatum [41,110], and combined GABA-glutamate effects have also been observed in the striatum [41]. However, the functional significance of this co-transmission in lampreys is still unknown.

Surprisingly, dopamine neurons in the lamprey have been found to differentially co-release these two neurotransmitters. As previously described, the same dopaminergic neurons have been found to target the striatum and tectum simultaneously [8,41]. However, these neurons co-release dopamine and glutamate only in the striatum, whereas only dopamine is released in the tectum. Although most of the neurons projecting to both structures co-express dopamine and glutamate, both glutamate and the dopamine marker TH are present in the axons targeting the striatum, whereas only TH is found in the fibers targeting the tectum. Accordingly, SNc/VTA stimulation results in both dopaminergic and glutamatergic effects in the striatum, but only in dopaminergic modulation in the tectum [41]. Although no studies have been carried out studying transmission in different branches of dopaminergic neurons that simultaneously target two structures, SNc/VTA neurons have been shown to release glutamate and dopamine at different sites within the same axon [103], suggesting that similar molecular mechanisms underlying the functional segregation of neurotransmitter release appeared early in vertebrate evolution and have been retained.

## 8. Evolution of Mesodiencephalic Dopaminergic Neurons

As discussed above, the available data strongly support that the diencephalic dopaminergic population of the lamprey is the homologue of the mammalian SNc/VTA. However, this homology has been somewhat controversial, as the SNc/VTA has been classically considered mesencephalic. Nevertheless, later studies in mammals have shown that SNc and VTA neurons originate in diencephalic, mesencephalic, and isthmic domains [111,112,113]. For this reason, these two populations, as well as the retrorubral field, are often referred to as the mesodiencephalic dopaminergic neurons. A similar situation can be observed in birds and reptiles, although dopaminergic neurons have a more restricted distribution [113,114]. In amphibians, the dopaminergic neurons projecting to the striatum are mostly located in the diencephalon, although they also occupy mesencephalic domains [115,116,117]. In elasmobranchs, dopaminergic neurons projecting to basal telencephalic regions form separated populations in the diencephalon and mesencephalon [118,119]. In teleosts, no dopaminergic neurons are present in the basal mesencephalon. This has led to a long debate regarding the presence of an SNc/VTA homologue, as the basal ganglia have not been characterized in this group and the findings disagree with the classic mesencephalic view. As in the lamprey, dopaminergic neurons that project to the striatum are located in the posterior tuberculum, and this population has therefore been proposed to be the homologue of the mammalian SNc/VTA [21,113,120,121]. On the contrary, studies using molecular marker expression have suggested that some of the neurons in the PT are the homologue of the A11 mammalian population that projects to the spinal cord. This hypothesis is based on them requiring Nkx2.1 and Otp to differentiate, which is true also for mammals [88,109,113], and also because descending neurons in the posterior tuberculum have been implicated in feedback regulation of sensory systems [122]. Recent studies have shown that zebrafish dopaminergic populations of the PT express shh, and that these are the only dopaminergic neurons to do so [123]. In mammals, all divisions comprising the mesodiencephalic dopaminergic neurons that correspond to the SNc/VTA arise exclusively from shh-expressing cells [124]. These results thus reinforce the homology of the zebrafish neurons in the PT with the mammalian SNc/VTA [123,125]. Additionally, the Nurr1 gene, which is mandatory for the differentiation of midbrain dopaminergic neurons in mammals, is also expressed in the zebrafish PT [126]. Altogether, it seems most likely that homologue populations to the mammalian SNc/VTA and A11 groups are present in the teleost PT [127].

A segmental approach therefore clarifies the diencephalic position of the lamprey population, while the multimeric nature of the SNc/VTA explains how, from an ancestral dopaminergic population that appeared in early vertebrates, several scenarios with small differences have evolved. The diencephalic portion of the SNc/VTA innervating the striatum is present in all vertebrate groups, indicating that it was the earliest to evolve and remains largely conserved. The presence of a mesencephalic population in elasmobranchs indicates that this component appeared early after the cyclostome–gnathostome split, and the lack of mesencephalic dopaminergic neurons in teleosts suggests that this component was lost secondarily in this lineage [113]. Lungfishes, which belong to the group of sarcopterygian fishes and are therefore placed in the sister branch of actinopterygian fishes (which includes teleosts), have a dopaminergic population in the mesencephalic tegmentum comparable to the SNc/VTA mesencephalic portion of mammals [127,128]. Interestingly, structures with multiple similarities to the mammalian basal ganglia have been identified in arthropods [129], representing a separate evolutionary branch to that of vertebrates. Although more studies are needed to corroborate these similarities and to elucidate the evolutionary origin of the basal ganglia and its modulation from the SNc/VTA, this suggests that their origin might be much older than the appearance of vertebrates and may relate to a common ancestor, perhaps an annelid worm. Although data are still scarce, there is evidence for dopaminergic neurons in the mesodiencephalic primordium of amphioxus, suggesting that the basic blueprint of the SNc/VTA was already present in non-vertebrate chordates [130,131]. Altogether, it seems that both the basal ganglia and the SNc/VTA already evolved in invertebrates, and the basic structure was largely conserved in evolution, although expanding in complexity in the vertebrate lineage.

## 9. Evolution of the Mesolimbic and Mesocortical Pathways

In mammals, the striatum is divided into a ventral striatum, also known as the nucleus accumbens, and a dorsal striatum. The nucleus accumbens receives inputs from limbic areas and the hippocampus, and its dopaminergic innervation arises in the VTA, via the so-called mesolimbic pathway [132]. The dorsal striatum is in turn divided into the dorsolateral or somatomotor part and the dorsomedial part, with input from the prefrontal areas, and the former receives dopamine input mostly from the SNc via the nigrostriatal pathway. The mesocortical pathway from the VTA to cortical areas completes the main dopaminergic pathways from mesodiencephalic neurons (see [5]). In lampreys, no differentiation between ventral and dorsal striatum has been so far identified [36], and there is currently no evidence for the SNc/VTA exhibiting dopaminergic projections to pallial regions (cortical homologue) [3], although more experiments are still warranted. Similarly, there is no evidence for projections from the dopaminergic neurons in the PT to pallial areas in zebrafish [88,113]. Thus, it seems that the nigrostriatal pathway was the first set of projections to appear in vertebrates, and that the mesocortical and mesolimbic pathways may have evolved later. However, it is important to note that further studies are needed, given that data concerning other basal vertebrates remains sparse. In amphibians, the organization of the dopaminergic innervation of the striatum shows a high degree of similarity with mammals, and as the three pathways are present, their appearance precedes the split of tetrapods [115,116,117], although when and how this occurred remain unknown.

We have so far discussed the nigrostriatal pathway and referred to the lamprey dopaminergic population as the homologue to the SNc/VTA. The lack of segregation between the dorsal and ventral striatum in these animals, as well as the absence of proven mesocortical and mesolimbic pathways, constrains a clear separation between SNc and VTA. In terms of projections, the lamprey dopaminergic population in the posterior tuberculum could therefore be considered a homologue of the SNc. However, apart from the three pathways mentioned above, the overall SNc and VTA connectome is quite similar in mammals [133], and further studies are warranted to establish whether the lamprey lacks mesocortical and mesolimbic pathways. The presence of a VTA component in lampreys should therefore not yet be excluded. In amphibians, neurons in the SNc and VTA are intermingled [117]. As a result, more studies are necessary to test whether a separate VTA-like population is also present in lampreys. Such studies would benefit from including a comparison of the transcription factors that differentiate the lamprey neurons from those known to specify the different mesodiencephalic groups in mammals [93,134], and also examining the inputs of these neurons to see if there are differences among subpopulations [133].

## 10. Conclusions

We can conclude that the SNc/VTA, together with its key role in modulating the basal ganglia, originated early in the vertebrate phylogenetic lineage and has retained a high degree of conservation. It is consequently clear that the appearance of the basal ganglia and their modulation from the SNc/VTA signified a big leap in evolution, providing new advantageous tools for survival. While lampreys command a limited behavioral repertoire compared to mammals, primarily limited to locomotion, steering, foraging, or control of body position, they nevertheless need to compete in a complex aquatic environment. Thus, it is not surprising that the basal ganglia and dopaminergic system were well-developed already in basal vertebrates, providing the basic machinery to signal salient events and evaluate actions, ultimately contributing to behavioral selection, motor control, and learning.

## Figures and Tables

**Figure 1 ijms-22-11284-f001:**
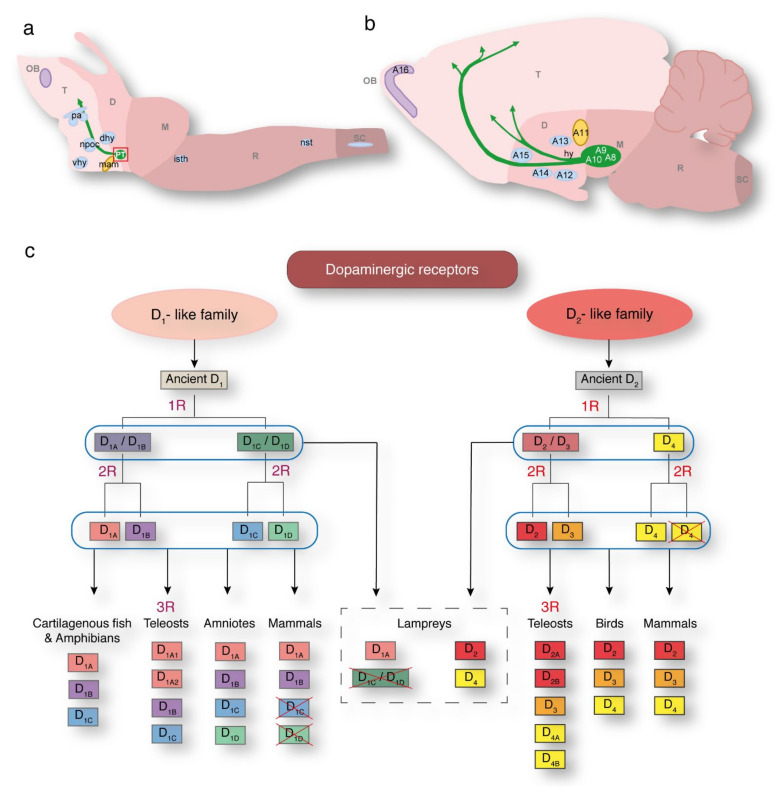
Distribution of dopaminergic neuronal cell groups in the brain of a basal vertebrate (lamprey, **a**) and a mammal (rat, **b**). In the lamprey, as in other vertebrates, dopaminergic cells are distributed in discrete populations from the olfactory bulb to the hypothalamus and rhombencephalon, as illustrated schematically in a sagittal view (**a**). The most rostral dopaminergic population in lamprey (purple) is located in the olfactory bulbs and would therefore correspond to the mammalian A16 group. In the preoptic area, there are CSF-contacting cells that resemble the suprachiasmatic nucleus population of other vertebrates [7]. There are also a few populations in the hypothalamus. The population located in the PT (green, and highlighted by a red square) is the homologue of the mammalian substantia nigra *pars compacta*/ventral tegmental area (SNc/VTA). Green arrows indicate the nigrostriatal projection in the lamprey brain, and the nigrostriatal, mesolimbic, and mesocortical pathways in the rat brain. The population in the mammillary area in lamprey (yellow) has been proposed to be the homologue of the mammalian A11 group due to its projections to the spinal cord [12]. Blue areas indicate dopaminergic populations whose homology has not yet been determined. Two populations in the lamprey rhombencephalon, located in the isthmic and nucleus of the solitary tract regions, and one more population in the spinal cord complete the lamprey dopaminergic groups. (**b**) Sagittal schematic of a rat brain showing the distribution of dopaminergic neurons from the olfactory bulbs to the mesencephalon. Modified from [11]. (**c**) Schematic representation of the phylogenetic evolution of dopaminergic receptors in vertebrates. The repertoire of D_1_-like and D_2_-like receptor subtypes present in the major vertebrate linages is shown at the bottom of the figure. The receptors present in each lineage are the result of the two WGDs that occurred in vertebrates (1R and 2R), and lineage specific gene losses (crossed out boxes). In the case of teleosts, an extra WGD (3R) took place that gave rise to extra subtypes. In the case of lampreys, the receptors present are likely the result of their divergence between both WGDs. However, it is also possible that lampreys diverged after both duplications took place, and that the extra copies that would have arisen were lost. OB, olfactory bulb; T, telencephalon, D; diencephalon; M, mesencephalon; R, rhombencephalon; SC, spinal cord; pa, preoptic area; dhy, dorsal hypothalamic nucleus; npoc, nucleus of the postoptic commisure; vhy, ventral hypothalamus; PT, posterior tuberculum; mam, mammillary area; isth, isthmus; nst, nucleus solitary tract; hy, hypothalamus.

**Figure 2 ijms-22-11284-f002:**
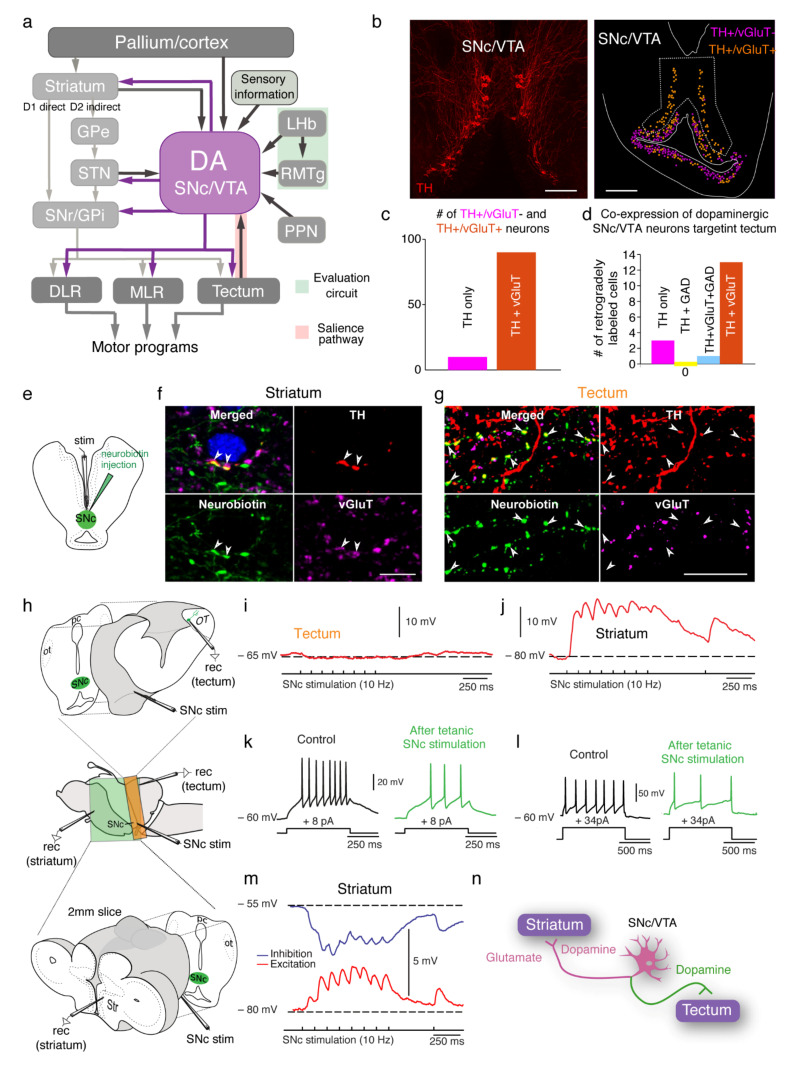
SNc/VTA connectivity in the lamprey is largely identical to that of mammals. Apart from projections to the striatum, the SNc/VTA directly projects to motor centers. (**a**) A schematic showing the overall connectivity of the lamprey SNc/VTA. The striatum, pallium (cortex homologue), lateral habenula, and pedunculopontine nucleus (PPN) provide inputs, as do different sensory regions. Among these sensory inputs, the tectum (homologue of the mammalian superior colliculus) provides salience information (shadowed in red). The lateral habenula provides direct inputs and regulates dopaminergic activity via indirect projections through the rostromedial tegmental nucleus (RMTg). This is the same evaluation circuit found in mammals (shadowed in green). Apart from the well-studied projection to the striatum, there are also direct projections to motor centers, both in lampreys and in other vertebrates, including mammals. DLR, diencephalic locomotor region; GPe, globus pallidus externa; MLR, mesencephalic locomotor region; SNc, substantia nigra *pars compacta*; SNr, substantia nigra *pars reticulata*; GPi, globus pallidus interna; STN, subthalamic nucleus; VTA, ventral tegmental area. Modified from [3]. (**b**) (right) Microphotography of the lamprey SNc/VTA dopaminergic population stained with an anti-TH antibody. Scale bar: 100 µm. (left) Mapping of neurons in the SNc/VTA (delimited with a dotted line) expressing only dopamine (pink) and co-expressing dopamine and glutamate (orange). Scale bar: 200 µm. The percentage of each type is shown in the graph in (**c**). (**d**) Graph showing the number for each of the neurotransmitter phenotypes exhibited by the neurons in the SNc/VTA that project to the tectum. TH is used as a marker for dopamine, and vGluT and GAD for glutamate and GABA, respectively. (**e**) Schematic drawing showing the injection site in the SNc/VTA that results in anterogradely labeled fibers in the striatum and tectum. (**f**) Anterogradely labeled fibers from injections in the SNc/VTA (green) express both TH (red) and vGluT (pink). Scale bar: 10 µm. (**g**) On the contrary, fibers targeting the tectum (green) express TH (red), but no vGluT (pink). Scale bar: 25 µm. (**h**) Experimental preparations used to record intracellularly in the tectum (OT; top) or striatum (str; bottom) in response to stimulation in the SNc/VTA. (**i**) No glutamatergic effects are evoked in tectum in response to SNc/VTA stimulation, whereas the same stimulation results in EPSPs in the striatum (**j**). ot, optic tract; pc, posterior commissure. (**k**) Tetanic stimulation of the SNc/VTA evokes dopaminergic effects in the tectum. A representative tectal neuron is shown that undergoes a reduction in excitability (therefore expressing the D_2_ dopamine receptor). For the same stimulus pulse, the number of evoked action potentials is lowered after the tetanic stimulation (green), in comparison to the control stimulation (black), in which the SNc/VTA tetanic stimulation has not yet been performed. (**l**) The same experiments show that SNc/VTA stimulation results in excitability changes in striatal neurons. (**m**) In some cases, both glutamatergic (EPSPs) and inhibitory (IPSPs) effects are observed in striatal neurons after SNc/VTA stimulation. (**n**) A summarizing schematic. Dopaminergic neurons in the lamprey SNc/VTA simultaneously target the striatum and tectum and co-express dopamine and glutamate. Both neurotransmitters are released in the striatum, whereas in the tectum, only dopamine is released. Modified from [41].

**Figure 3 ijms-22-11284-f003:**
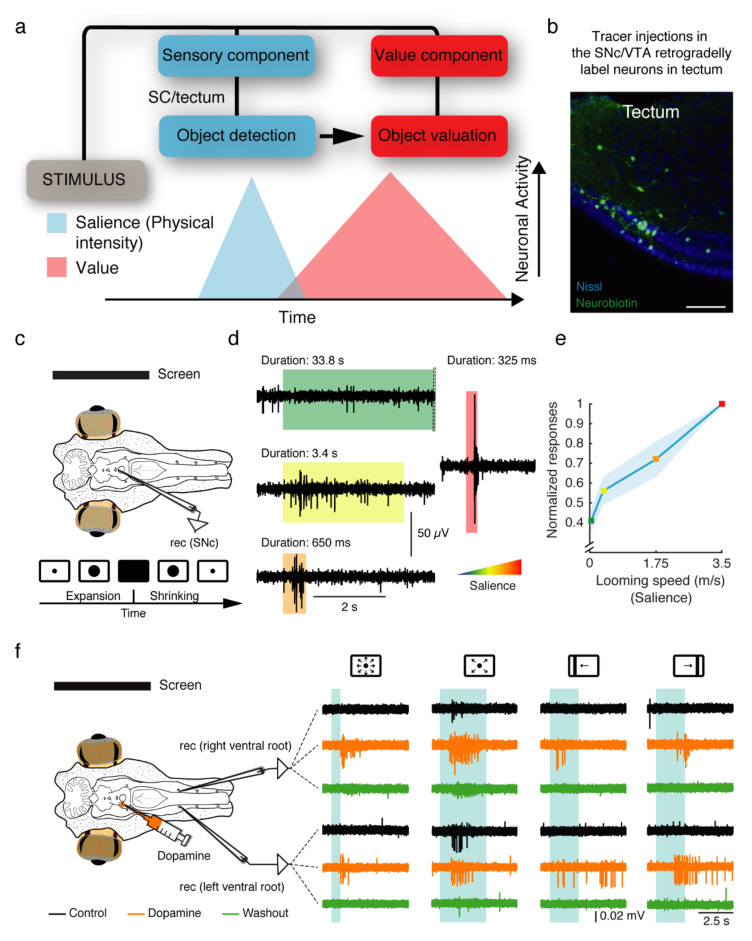
Two-component response of dopaminergic neurons. (**a**) Dopaminergic SN/VTA neurons show a two-component phasic response. The first component depends only on the salience (physical intensity) of the stimulus. The second component depends on the inferred value of the stimulus. The process of moving from salience/detection to object valuation has been proposed to take place via intermediate steps mediated by different brain areas. For visual stimuli, the SC/tectum has been proposed to provide the salience/detection information, whereas additional areas, including the cortex, are involved in the identification and evaluation of the stimulus. However, it is not yet fully understood where and how this processing takes place (modified from [59]). (**b**) Retrogradely labeled neurons in the tectum after a neurobiotin injection in the SNc/VTA indicate that the same pathway thought to convey salience information to the SNc/VTA in mammals is also present in lampreys. Scale bar: 100 µm. (**c**) The SNc/VTA in the lamprey signals visually salient responses. When looming stimuli (a dot expanding in the screen) are presented to an eye–brain preparation allowing visual stimuli to be presented while recording electrophysiological activity, the responses in the SNc/VTA increase in parallel to the speed of the stimulus. Representative traces at two different speeds are shown in (**d**). The shadowed area indicates the duration of the visual stimulation and its level of salience. (**e**) Graph showing the increase in SNc/VTA activity parallel to the increase in looming speeds. (**f**) The net effect of the dopaminergic innervation from the SNc/VTA to the tectum is an increase in the responsiveness to sensory stimuli. When visual stimuli (looming dots and moving bars) are presented to the eye–brain preparation described above, larger motor responses are evoked after dopamine is injected in the tectum. The traces show the activity in a rostral pair of ventral roots before (black) and after injecting dopamine in the tectum (orange). The effects are reversed after several minutes (green). Modified from [8].

## Data Availability

Not applicable.

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
