# Peer review of "The Dopaminergic Control of Movement-Evolutionary Considerations"

_ijms, 2021, doi:10.3390/ijms222011284_

Round 1

Reviewer 1 Report

The authors first review the literature about the dopaminergic neuronal populations and receptors in the lamprey. They then establish the notion that the basal ganglia are largely conserved from lampreys to mammals, and the SNc/VTA is present at the base of vertebrate evolution. They illustrate the early origin of the dopaminergic system and how it codes salience and reward, and the direct dopaminergic control of downstream brain centers. They further describe the co-release of dopamine and other neurotransmitters, and the evolution of mesodiencephalic dopaminergic neurons and the mesolimbic and mesocortical dopaminergic pathways. They conclude that the SNc/VTA evolves early in the vertebrate lineage and its role in modulating the basal ganglia has been highly conserved, providing the basic machinery to signal salient events and evaluate actions, ultimately contributing towards behavioral selection, motor control, and learning.

General Comments:

This review article is well written and provides great insights and informative figures.

Specific Comments:

Line 67 “…D2-like family receptors present introns…”

Replace “present” with “contain”.

Lines 278-288: Figure 3(b) “…Retrogradely neurons in the tectum after a neurobiotin injection in the SNc/VTA…”

Retrogradely labeled neurons…

Line 289: “Neurons in the deep layer subsequently project contra- or ipsilaterally to the brainstem…”

…contra-or ipsi-laterally…

Line 301: “…indicating that that the SC-SNc/VTA path…”

Delete the extra “that”.

Lines 460-462: “Interestingly, structures with multiple similarities with the mammalian basal ganglia have been identified in arthropods [129], representing a separate evolutionary branch to that of vertebrates.”

Arthropods are invertebrates. Please clarify the intention of this sentence.

Reviewer 2 Report

In their paper entitled "The dopaminergic control of movement-evolutionary considerations", the authors have provided an overview of existing research in basal vertebrates, focusing mainly on lampreys, which belong to the oldest group of living vertebrates. This review is complex and summarizes recent findings on dopaminergic control, dopamine, and neurotransmitters with an appropriate summary of the literature. The manuscript is well written and will be of interest to readers of the journal IJMS. I suggest to accept it for publication.

Author Response

We thank the reviewer for the kind comments